# Emission Factors for Biofuels and Coal Combustion in a Domestic Boiler of 18 kW

**Marianna Czaplicka \*, Ewelina Cieślik, Bogusław Komosiński and Tomasz Rachwał**

Institute of Environmental Engineering, Polish Academy of Sciences, 34 M. Skłodowskiej-Curie Street, 41-819 Zabrze, Poland; ewelina.cieslik@ipis.zabrze.pl (E.C.); boguslaw.komosinski@ipis.zabrze.pl (B.K.); tomasz.rachwal@ipis.zabrze.pl (T.R.)

**\*** Correspondence: marianna.czaplicka@ipis.zabrze.pl

**Abstract:** The differences in the pollutant emissions from the combustion of bituminous coal and biofuels (wood, straw, and miscanthus pellets) under real-world boiler operating conditions were investigated. The experiments were performed on an experimental installation that comprised an 18 kW boiler, used in domestic central heating systems, equipped with a retort furnace, an automatic fuel feeder, a combustion air fan, and a fuel storage bin. The emission factors of gaseous pollutants, particulate matter, organic carbon, elemental carbon, and polycyclic aromatic hydrocarbons (PAHs), as well as some PAH concentration ratios for coal and biofuel combustion, were determined. The obtained results indicate that fuel properties have a strong influence on the emission factors of gaseous and carbonaceous pollutants. The total particulate matter (PM) emissions from the biofuel combustion were about 5-fold lower than those from the coal burned in the same boiler. The emission factors of the total carbons from the biofuel combustion were between 10 and 20 times lower than those from the coal combustion. The mean organic carbon (OC) and elemental carbon (EC) emission factors, based on the burned fuel, were 161–232 and 42–221 mg/kg for the biofuels and 1264 and 3410 g/kg for the coal, respectively. The obtained results indicate that molecular diagnostic ratios, based on the concentration of PAHs, vary significantly, depending on the fuel type.

**Keywords:** combustion; coal; biofuels; emission; PAH molecular diagnostic ratios

## 1. Introduction

Biofuel combustion emits a substantial amount of CO, which can contribute to the $CO_2$ level through oxidation [1,2]. Forest and agriculture solid biomass fuels are used as the main or additional fuel in power plants with boilers of various types and in households as a traditional bioenergy. Biomass fuels, mainly wood and plant residues, are important sources of primary energy and account for about 13% of global fuel consumption [3]. Biomass is a complex heterogeneous mixture of organic and inorganic matter, containing various associated phases or minerals [4]. It is well known that, during combustion, the structural organic compounds in biomass play a role similar to lithotypes in coal [5]. The emission factors (EF) for coal and biofuels under specific combustion conditions have been presented in numerous papers [6–9]. The emissions from the combustion and composition of the organic compounds released from biomass burning depend on, among other things, the fuel composition, combustion rate, and boiler type [10,11]. Reductions in the total CO, organic carbon (OC), elemental carbon (EC), particulate matter (PM), and polycyclic aromatic hydrocarbon (PAH) emissions of 95%, 98%, 98%, 98%, 98%, 88%, 88% and 71%, respectively, can be achieved by replacing raw biomass fuels, burned with pellets in traditional kitchen stoves, with biofuels burned in modern pellet burners [12]. The mass concentrations of particles in the flue gas from small-scale combustion appliances vary from 7 to 67 mg/m³ (at 13% $O_2$) [13]; 50 mg/m³ (at 11% $O_2$); 5000 mg/m³ for a village wood cook stove; 20–100 mg/m³ in the case of a residential boiler (2–10

kW) for wood chips or miscanthus; and 20 mg/m$^3$ for pellet stoves with a fixed grate of 2–25 kW [14]. Soot accounts for around 30% of the total PM10 in biofuel combustion in manually fired systems, and organic carbon accounts for 30–40% [15].

The EC/OC, EC/PM, and OC/PM ratios are often used as indicators of the contribution of energy sources and to assess the impact of PM derived from the combustion of fossil fuels on climate change. The combustion of biofuels produces organic compounds, which form as a consequence of the pyrolysis of lignin and cellulose, including carbonyl compounds, saccharides, phenol and derivatives, methoxylated phenolic compounds, volatile organic compounds (VOCs), and polycyclic aromatic hydrocarbons [16,17]. Venkataraman et al. [18] reported that the emission factors of the total PAH from wood, briquette, and dung-cake combustion ranged from 2.0 to 3.2 mg/kg, 2.8 to 3.0 mg/ kg, and 3.1 to 5.5 mg/ kg, respectively.

The objectives of this study were to determine the emission factors of CO, $CO_2$, $SO_2$ particulate matter, organic carbon, elemental carbon, and total organic matter from the burning of three types of pellets and bituminous coal in a commercial boiler. The diagnostic ratios of the PAHs of the fuels studied were also characterized, and the OC/total organic matter (TC) and TC/PM ratios were determined.

## 2. Experiments

### 2.1. Fuel Characteristics

Three biomass pellets made of coniferous wood, i.e., pine and spruce wood, cereal straw pellets, miscanthus pellets, and bituminous coal, were combusted in a domestic central heating (CH) boiler of 18 kW. The pellets were 5–35 mm in length and 6–10 mm in diameter. Proximate, ultimate, and moisture analyses were conducted, according to the standard European methods [19–22]. The moisture content and ash content were determined using the gravimetric method. The total carbon, hydrogen, nitrogen, and sulfur contents were determined by high-temperature combustion using IR detection (CHS 900 firmy Eltra GmbH t). The chlorine content was determined by titration. The oxygen content was calculated. The calorimetric method was used to determine the heating value of the fuels.

### 2.2. Boiler Descriptions

The combustion experiments were conducted in a typical domestic central heating boiler of 18 kW, with an efficiency of 85%, a retort furnace, an automatic fuel feeder, a combustion air fan, a fuel storage bin, an external heat exchanger for heat recuperation, smoke and particulate sampling points, and a control and measurement system consisting of a gravimetric dust meter, dust aspirator, S-tube, and exhaust gas analyzer, DX-4000 Gasmet. The boiler operated with a variable power, controlled by the value of the return water temperature from the external heat exchanger. The boiler was certified for solid fuel combustion. The technical specifications of the boiler are presented in Table 1.

**Table 1.** The technical specifications of the boiler.

| Parameter | Unit | Value |
|---|---|---|
| Nominal heating output | kW | 18 |
| Heating surface | m$^2$ | 1.7 |
| Efficiency | % | 85.0–85.9 |
| Maximum water temperature | °C | 95 |
| Maximum operating pressure | MPa | 0.1 |

The boiler operated with a variable power, controlled by the value of the return water temperature from the external heat exchanger. The combustion conditions of the fuels are given in Table 2. The combustion experiments were conducted in triplicate for each fuel.

**Table 2.** The combustion conditions.

| Parameter | Unit | Fuel | | | |
|---|---|---|---|---|---|
| | | Bituminous Coal | Pelletized Coniferous Wood | Pelletized Cereal Straw | Pelletized Miscanthus |
| Flue gas temperature | °C | 112 | 87 | 104 | 120 |
| Flue gas volume flow (measurement conditions) | m³/h | 122 | 115 | 122 | 111 |
| Moisture content in flue gases | kg/kg | 0.050 | 0.041 | 0.045 | 0.050 |
| Oxygen share in the dry flue gases | % | 16.1 | 15.5 | 18.8 | 16.8 |
| $CO_2$ share in the dry flue gases | % | 5.4 | 6.4 | 6.8 | 7.5 |
| Amount of fuel burned | kg | 18.82 | 26.86 | 28.00 | 37.59 |
| Boiler operation time | h | 8.1 | 7.4 | 6.9 | 10.4 |
| Fuel consumption per unit of time | kg/h | 2.33 | 3.63 | 4.06 | 3.60 |

### 2.3. Sample Collection and Measurement

During the combustion studies, some measurements and analyses were carried out, and samples of gases and dust were collected for laboratory analysis. The gas velocity was measured at 1-min intervals by determining the dynamic pressure at the measurement points. The $CO_2$, CO, and $SO_2$ concentrations in the flue gas stream were measured (exhaust gas analyzer, DX-4000 Gasmet). During the combustion experiments, the $CO_2$, CO, and $SO_2$ concentrations in the flue gas stream were measured. The expanded uncertainty, $U_i$, of the measurement of the concentrations of gaseous pollutants and dust is given in Table 3.

**Table 3.** Extended uncertainty of the measurement of gaseous components and particulate matter (PM).

| Compound | Extended Uncertainty of the Measurement, % |
|---|---|
| CO | 7.3 |
| $SO_2$ | 10.5 |
| $CO_2$ | 6.2 |
| $O_2$ | 4.4 |
| PM | 22.0 |

Determinations of the particulate matter concentration and mass flow, as well as the physicochemical parameters of the flue gases, were conducted, according to the standard ISO [23]. A sampling port in the flue was on a side stream, separated from the mainstream under isokinetic conditions. Samples of the pollutants were collected using a titanium test probe and Whatman glass microfiber filters (grade: GF/A).

### 2.4. Analytical Methods

The PM mass was measured by weighing the filters, before and after sampling, using a 0.00001 g digital balance. The EC and OC content in PM were analyzed using a Sunset EC/OC analyzer (Sunset, USA), according to the procedure described in [24].

The concentration of PAHs, e.g., phenanthrene, anthracene, fluoranthene, pyrene, benzo[a]anthracene, chrysene, benzo[b]fluoranthene, benzo[k]fluoranthene, benzo[a]pyrene, indeno[1,2,3-cd]pyrene, and benzo[g,h,i]perylene, in the flue gas were determined. Filters with the

addition of an internal standard—deuterated PAHs—were extracted with dichloromethane. The extraction was carried out in an ultrasonic bath 2 times for 30 min each time. Quantitative analysis was performed using GC MS (QP-2010 Plus Shimadzu), equipped with a ZB-5MS column (30 m × 0.25 mm × 0.25 μm). The analyses were carried out using a programmed increase in the operating temperature within the range of 80–280 °C. The initial temperature of the analysis was 80 °C, which was maintained for 1 min, then 280 °C was reached at an 8 °C/min rate and maintained for 9 min. The total analysis time was 40 min. The expanded uncertainty of the assay was estimated to be 17% (with a confidence level of 95% and a coverage factor $k$ = 2).

The recoveries obtained for the reference samples spiked with deuterated PAHs ranged from 77 to 105%.

## 3. Results and Discussion

The physical–chemical properties of the biofuel pellets and coal are shown in Table 4.

**Table 4.** Physical–chemical properties of the fuels (average ± standard deviation (SD), *n* = 3).

| Parameters | Bituminous Coal | Pelletized Coniferous Wood | Pelletized Cereal Straw | Pelletized Miscanthus |
|---|---|---|---|---|
| Moisture, % | 5.32 | 4.02 | 3.91 | 3.45 |
| Ash content, % | 8.71 | 0.37 | 6.64 | 2.06 |
| Volatile matter (VM), % | 33.94 | 81.72 | 72.63 | 78.73 |
| Lower heating value (LHV), MJ/kg | 26.95 | 17.43 | 15.33 | 15.73 |
| **Elemental analysis, %** | | | | |
| C | 69.21 ± 0.03 | 50.11 ± 0.13 | 44.98 ± 0.10 | 47.96 ± 0.10 |
| H | 4.91 ± 0.00 | 7.25 ± 0.01 | 6.65 ± 0.00 | 7.12 ± 0.03 |
| N | 1.19 ± 0.00 | 0.10 ± 0.01 | 0.62 ± 0.03 | 0.23 ± 0.02 |
| S | 0.81 ± 0.01 | <0.03 | 0.08 ± 0.01 | 0.04 ± 0.01 |
| Cl | 0.28 ± 0.03 | <0.10 | 0.11 ± 0.01 | 0.13 ± 0.03 |
| O | 9.57 | 28.60 | 37.56 | 39.53 |

The moisture content of the coniferous wood, cereal straw, and miscanthus pellets was on a similar level of 4.02, 3.91, and 3.45 %, respectively. The coal was characterized by a moisture level of 5.32. The pellets were characterized by a high content of volatile matter in comparison to that of coal. The volatile matter (VM) of the pellets ranged between 72 and 82% wt. Coal had a higher ash content than the pellets. The ash content of the pellets was 6.64, 2.06, and 0.37% for cereal straw, miscanthus, and coniferous wood, respectively. The lower heating value (LHV) for coal was higher than that for biofuels and was 27 MJ/kg. For the pellets from biomass, the LHV ranged from 17.4 (coniferous wood) to 15.3–15.7 MJ/kg for cereal straw and miscanthus, respectively. Elementary analysis showed that coal was characterized by a higher content of carbon, nitrogen, and chlorine and a lower content of hydrogen in comparison to biofuels. The chemical composition of biofuels and coal is similar to that reported in other studies [6,11].

### 3.1. Emission Factors

The emission factors were calculated by dividing the pollutant mass flow by a fuel consumption per unit of time [25]. The fuel consumption per unit of time was calculated by dividing the amount of fuel burned (the fuel in the fuel feeder was weighed before and after combustion) by the boiler operation time. The emission factors (EFs) for basic gaseous pollutants, i.e., $CO_2$, CO, and $SO_2$, are presented in Table 5.

**Table 5.** Emission factors (EFs) for gaseous pollutants, gram per kilogram (g/kg) fuel.

| Fuel | CO | $CO_2$ | $SO_2$ |
|---|---|---|---|
| Bituminous coal | 6.7 | 1 573 | 5.1 |
| Pelletized coniferous wood | 61.8 | 2 753 | 0.3 |
| Pelletized cereal straw | 16.8 | 725 | 0.5 |
| Pelletized miscanthus | 56.0 | 1 302 | 0.6 |

As for the biofuel combustion, the lowest CO emission was observed for the cereal straw combustion and amounted to 16.8 g/kg, while for the miscanthus and coniferous wood combustion, it amounted to 56 g/kg and 62 g/kg, respectively. On the other hand, the CO emissions during coal combustion was at a level of 6.7 g/kg. The $CO/CO_2$ ratio for the coniferous wood and cereal straw pellets was on a level of 0.03. However, that for the pelletized miscanthus was 2 times higher. In the case of coal combustion, the $CO/CO_2$ ratio was 0.01 and was lower than the values obtained for biofuels. This result can be explained by the different fuel oxidation efficiency in the boiler. A similar relationship has been demonstrated by Verma et al. [26]. As for the $SO_2$ emissions, the three biofuels were similar and ranged between 0.3 and 0.6 mg/kg, whereas coal achieved a value of 5.1 mg/kg. It is well known that the emission factor for $SO_2$ depends on the sulphur content in fuel. In the presented studies, the sulphur content in biofuels was lower than that in coal (0.08–0.4% and 0.81% for biofuels and coal, respectively).

The PM, TC, OC, and EC emission factors, obtained during the combustion of biofuels and coal, are presented in Table 6. For the biofuel combustion, the average PM emission factors ranged from 1.7 g/kg (coniferous wood pellets) to 2.7 g/kg (cereal straw pellets) and were lower than the emissions from the coal combustion, which amounted to 10.2 g/kg. It should be noted that the average PM emissions during the combustion of biofuels was about 5 times lower than that during the combustion of coal in the same boiler.

**Table 6.** EFs, PM (g/kg), organic carbon (OC), elemental carbon (EC), and total organic matter (TC) (mg/kg fuel).

| Fuel | PM | OC | EC | TC |
|---|---|---|---|---|
| Bituminous coal | 10.20 | 1 264 ± 1.23 | 3410 ± 1.53 | 4 674 |
| Pelletized coniferous wood | 1.65 | 161 ± 1.03 | 41.9 ± 0.95 | 202.9 |
| Pelletized cereal straw | 2.74 | 127.2 ± 0.45 | 120.7 ± 0.83 | 247.9 |
| Pelletized miscanthus | 2.06 | 232.4 ± 0.89 | 221.9 ± 0.54 | 454.3 |

It is known that three types of particles are mainly emitted during combustion, i.e., soot, organic particles, and inorganic particles. Soot and organic particles are the result of the incomplete combustion of fuels. As in the case of CO, the emission of organic particles depends on the combustion efficiency [26]. The PM emitted from combustion processes contains both elemental (EC) and organic (OC) carbon. The total carbon content (TC) in the exhaust gas is defined as the sum of OC and EC. The obtained results indicate that the emission factors of TC ($EF_{TC}$) from the combustion of biofuels ranged from 203 to 454 mg/kg, and they were much lower than those from coal combustion. In the case of coal combustion, the $EF_{TC}$ was 4,674 mg/kg (Table 6).

The research results indicate that the OC emission factor ($EF_{OC}$) ranges from 127 to 232 mg/kg in the case of pellet combustion, while for coal, it is 1,265 g/kg. In the case of coal combustion, the EC emission factor ($EF_{EC}$) was higher than the $EF_{OC}$. A different correlation was observed for pelletized coniferous wood. On the other hand, the $EF_{OC}$ and $EF_{EC}$ emissions were comparable to those for pelletized cereal straw and miscanthus.

The measured $EF_{OC}$ and $EF_{EC}$ for the pelletized coniferous wood are at a similar level to those reported by Shen et al. [15] for residential wood combustion in a typical cooking stove.

The contents of OC and EC in PM differently depended on the fuel type (Figure 1). The EC mass fractions in the particles emitted during coal combustion (33.3 wt%) were higher than those emitted during biofuel combustion (2.4–10.7 wt %), while the opposite was observed for OC (12.4 wt% and 4.4–11.2 wt% for the coal and biofuels, respectively). The mass concentrations of OC in PM were

very similar for coal (12 %), pelletized miscanthus (11%) and pelletized coniferous wood (10%), whereas for pelletized cereal straw, the OC content in PM was at a level of 4.4 wt %. This phenomenon can be explained by the properties of biofuels, since herbaceous plants, such as straw, have a higher burning rate than ligneous plants (pine and pellet fuels).

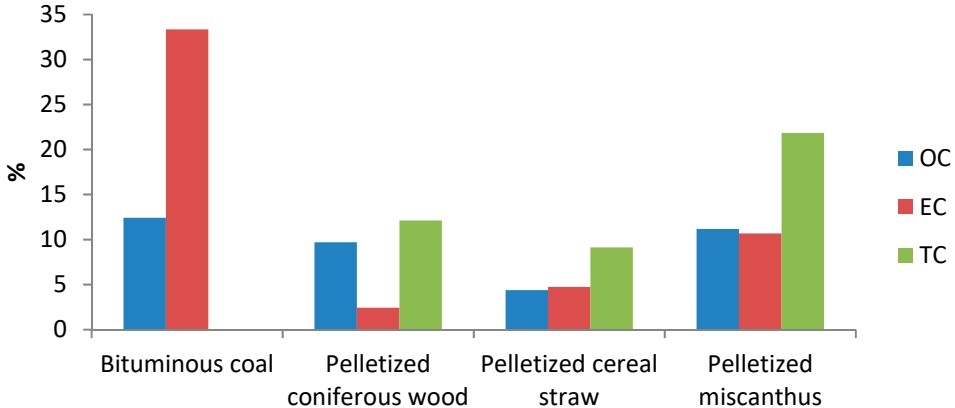

**Figure 1.** Mass fractions of TC, OC, and EC in PM, %.

The OC/TC and TC/PM ratios are useful indicators in the identification of the source emission for carbonaceous PM [27,28]. The determined OC/TC ratios were as follows: 0.8, 0.4, and 0.5 for coniferous wood pellets, cereal straw pellets, miscanthus pellets, respectively. In the case of miscanthus pellets, the OC/TC ratio was similar to that for miscanthus pellets, as was reported in the literature ($0.52 \pm 0.26$) [29]. In the case of coal combustion, the OC/TC ratio amounted to 0.3.

The OC/PM ratios for coal, pelletized coniferous wood, and pelletized miscanthus were at a level of 0.1–0.12 and were higher than those for pelletized cereal straw (0.04). The results indicate that the OC/EC ratios for biomass burning are higher than those for coal (Figure 2). A similar correlation was observed by Novakov et al. [30].

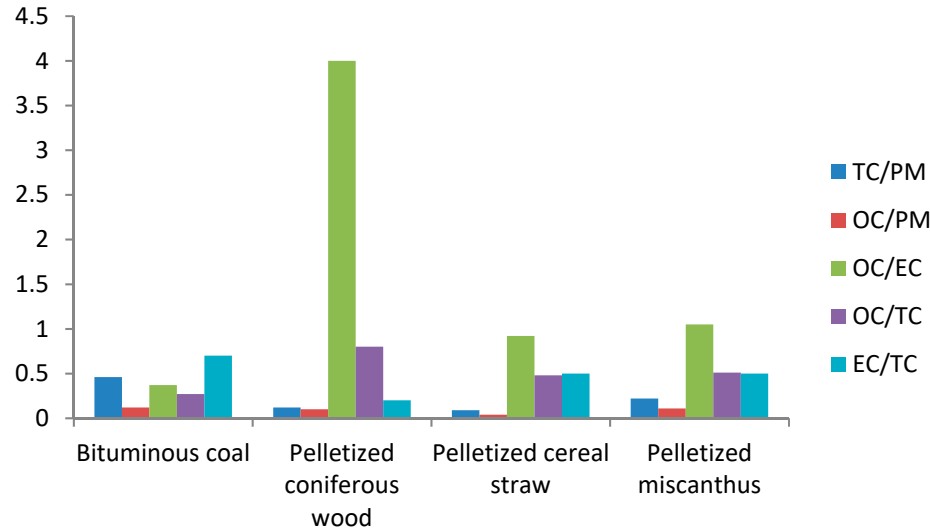

**Figure 2.** OC, EC, and PM mass ratios, derived for different fuel types.

In the present study, the OC/EC ratio ranged from 0.9 to 4.0 for biofuels. A lower OC/EC ratio has been reported for pelletized cereal straw (0.9) and pelletized miscanthus (1.1) in comparison to pelletized coniferous wood combustion (4.0) (Figure 2). A similar correlation was observed by Fernandes et al. [31], who presented an average OC/EC ratio of 0.85 for softwood combustion and

between 3.14 and 4.39 for hardwood combustion in a woodstove. On the other hand, Zhang et al. [32] provided an OC/EC ratio of 2.8 ± 1.3 for burning wood in a similar furnace.

Some studies have shown EC/TC ratios of between 0.01 to 0.11 for fuel combustion [33–35]. Bølling et al. [35] reported that the EC/TC ratio for incomplete combustion ranges from 0.5 to 0.8.

*3.2. Molecular Diagnostic Ratios*

The molecular diagnostic ratios of the defined pairs of individual compounds have often been used as tracers of different source categories of polycyclic aromatic hydrocarbons [36–39]. During combustion, PAHs may be formed from organic matter under oxygen-deficient conditions. Pyrosynthesis and pyrolysis are the two main mechanisms that can explain the formation of PAHs. It is well known that the mechanism formation of PAH during combustion includes: (1) Radical reactions, (2) Diels–Alder condensations, and (3) an ionic reaction mechanism. The thermodynamics of combustion favor the mechanism of radical formation. The emission factors of PAHs (EF$_{PAH}$) are strongly dependent on the properties of the fuel, the type of the furnace, and the combustion conditions. Venkataraman et al. [18] reported that the emission factors of the total PAH from wood, briquette, and dung-cake combustion range from 2.0 to 3.2 mg/kg, 2.8 to 3.0 mg/ kg, and 3.1 to 5.5 mg/ kg, respectively. PAHs and CO are both products of inefficient combustion and are therefore dependent on the temperature of combustion [40]. Levendis et al. [41] found a relationship between the CO and PAHs from the combustion of coal and waste tires. On the other hand, Rantanen [42] and Saez et al. [43] confirmed higher PAH emissions under inefficient combustion conditions, but did not find a correlation between the PAH and CO levels. Several authors [44–46] used molecular diagnostic ratios (MDRs) as an indicator of the distribution of PAH sources in the environment, especially in the air. For example, the ratio of ANT/(ANT+PHE) has been suggested as an indicator of petrogenics for pyrogenic sources. This ratio has been criticized in the past [46,47], because ANT is more reactive than PHE, and their environmental behavior is therefore very different. The BaA/(BaA+CHR) ratio allows one to discriminate between the same sources as those in the ANT/(ANT+PHE) ratio, but it is more representative. Kim et al. [48] pointed out that a value below 0.20 suggests petrogenic emissions, while a value >0.35 indicates combustion (pyrogenic emissions). The FLT/(FLT+PYR) ratio distinguishes between petro- and pyrogenic sources (<0.40 and >0.40, respectively), but it can also indicate whether the pyrogenic emissions result from fuel combustion (0.4–0.5) or from the combustion of other materials (>0.50). The IP/(IP+BgP) ratio is to be interpreted in a similar way to FLT/(FLT+PYR). Several authors [49–51] showed that an IP/(IP+BgP) ratio >0.5 indicates that the PAHs probably originated in the emissions from coal combustion. In Table 7, the diagnostic ratios of the PAHs, used as the source indicator of fuel combustion, as found in previous studies, are presented.

**Table 7.** Polycyclic aromatic hydrocarbon (PAH) ratios for the other fuels [52].

| Molecular Ratio | Coal | Crop Residue Pellets (Peanut Hull) | Coal Crop Residue (Peanut) | Wood | Pine Wood | Corn Straw |
|---|---|---|---|---|---|---|
| FLT/(FLT+PYR) | 0.57 | 0.51 | 0.5 | 0.5 | 0.54 | 0.53 |
| BaA/(BaA+CHR) | 0.51 | 0.37 | 0.3 | 0.4 | 0.45 | 0.46 |
| IP/(IP+BgP) | 0.61 | 0.62 | 0.5 | 0.5 | 0.54 | 0.53 |
| BbF/(BbF +BkF) | 0.67 | 0.65 | 0.5 | 0.6 | 0.49 | 0.52 |
| BaP/(BaP+CHR) | 0.45 | 0.56 | 0.5 | 0.5 | No | No |

No: Lack of a data.

On the basis of the PAH concentrations in flue gases, MDR indices for the tested fuels were calculated. The relationship between ANT/(ANT+PHE), FLT/(FLT+PYR), BaA/(BaA+CHR), IP/(IP+BgP), BbF/(BbF +BkF), and BaP/(BaP+BgP) are presented in Table 8.

**Table 8.** Polycyclic aromatic hydrocarbon diagnostic ratios for the investigation fuels.

| Molecular Ratio * | Bituminous Coal | Pelletized Coniferous Wood | Pelletized Cereal Straw | Pelletized Miscanthus |
|---|---|---|---|---|
| ANT/ANT+PHE | 0.10 ± 0.02 | 0.23 ± 0.03 | 0.07 ± 0.03 | 0.22 ± 0.02 |
| FLT/(FLT+PYR) | 0.60 ± 0.03 | 0.49 ± 0.02 | 0.55 ± 0.03 | 0.50 ± 0.03 |
| BaA/(BaA+CHR) | 0.50 ± 0.12 | 0.41 ± 0.13 | 0.36 ± 0.10 | 0.47 ± 0.09 |
| BbF/(BbF +BkF) | 0.58 ± 0.03 | 0.47 ± 0.05 | 0.53 ± 0.04 | 0.47 ± 0.05 |
| BaP/(BaP +CHR) | 0.45 ± 0.09 | 0.44 ± 0.03 | 0.42 ± 0.03 | 0.59 ± 0.13 |
| IP/(IP+BgP) | 0.53 ± 0.13 | 0.51 ± 0.10 | 0.50 ± 0.12 | 0.57± 0.09 |

* Ratio calculated by $n = 5$.

The obtained results indicate that the ANT/(ANT+PHE) ratio is very different, depending on the fuel type. For pelletized coniferous wood and pelletized miscanthusm, the ratio is about 0.22 and is higher than that for pelletized cereal straw and coal, which are 0.07 and 0.1, respectively. The FLT/(FLT+PYR) ratio ranges from 0.49 (pelletized coniferous wood) to 0.6 (coal) and is similar to the results presented by Shen et al. [53] and Cheruiyot et al. [54]. A higher BaA/(BaA+CHR) ratio was observed for coal in comparison to that for biofuels. This ratio was 0.5 for coal and 0.41, 0.36, and 0.47 in the case pelletized coniferous wood, pelletized cereal straw, and pelletized miscanthus, respectively. The BbF/(BbF+BkF) ratio for pelletized coniferous wood and pelletized miscanthus was at the same level (0.47), while for hard coal, it was 0.58. In the case of BaP/(BaP+CHR) ratios, a similar level of 0.4–0.45 was achieved for coal, pelletized coniferous wood, and cereal straw, while the ratio was lower than for pelletized miscanthus. For the IP/(IP+BgP) ratio, a similar correlation was observed. This ratio was at a level of 0.5–0.53 and 0.57 for coal, pelletized coniferous wood or cereal straw and pelletized miscanthus, respectively. The determined MDR are similar to the results presented in other publications despite different combustion conditions and physicochemical properties of biofuels. This similarity confirms their usefulness for identification of combustion sources.

## 4. Conclusions

The presented results showed that:

- The emissions of PM from the biofuel combustion were about 5-fold lower than those from the coal burned in the same boiler.

- For the biofuel combustion, the average PM emission factors ranged from 1.7 to 2.7 g/kg (cereal straw pellets) and were lower than the emissions from the coal combustion.

- The CO emissions during coal combustion were at a level of 6.7 g/kg and were the lowest in comparison with those from biofuel combustion, which ranged from 16.8 g/kg for cereal straw to 62 g/kg for pellet coniferous wood combustion.

- The emission factors of the total carbons from the biofuel combustion ranged from 9.1 to 21.8 mg/kg and were lower than those from the coal combustion.

- The OC/TC ratios substantially differed for the different biomasses, amounting to 0.8, 0.4, and 0.51 for coniferous wood pellets, cereal straw pellets, and miscanthus pellets, respectively, and were higher compared to those for the coal burning.

- The OC/EC ratios ranged from 1.1 to 4.0 for the biofuels and amounted to 0.4 for the coal combustion.

- The molecular diagnostic ratios for the ANT/(ANT+PHE) ratio substantially differed, depending on the fuel type. The FLT/(FLT+PYR) ratios ranged from 0.49 (pelletized coniferous wood) to 0.6 (coal). The BbF/(BbF +BkF) ratios for the pelletized coniferous wood and the pelletized miscanthus were at the same level. The BaP/(BaP +CHR) and IP/(IP+BgP) ratios appeared to be at a similar level for coal, pelletized coniferous wood, and cereal straw and were lower than those for pelletized miscanthus.

**Abbreviations, Symbols, and Acronyms**

EF: Emission factors
EC: Elemental carbon
LHV: Lower heating value
MDR: Molecular diagnostic ratios
OC: Organic carbon
PM: Particulate matter
PAHs: Polycyclic aromatic hydrocarbons
PHE: Phenanthrene
ANT: Anthracene
FLT: Fluoranthene
PYR: Pyrene
BaA: Benzo[a]anthracene
CHR: Chrysene
BbF: Benzo[b]fluoranthene
BkF: Benzo[k]fluoranthene
BaP: Benzo[a]pyrene
IP: Indeno[1,2,3-cd]pyrene
BgP: Benzo[g,h,i]perylene
TC: Total organic matter
VM: Volatile matter

**Author Contributions:** E.C. and M.C. conceived and designed the experiments; T.R. and B.K. performed the experiments; B.K. and E.C. analyzed the data; and M.C. wrote the paper.

**Funding:** Data analysis was supported by the Institute of Environmental Engineering of the Polish Academy of Sciences basic (statutory), research project no. A1/105/2015.

**Acknowledgments:** The authors thank Prof. J. Konieczynski for his comments and Dr. B. Mathews for his analyses of organic carbon.

**Conflicts of Interest:** The authors declare no conflicts of interest. The founding sponsors had no role in the design of the study; in the collection, analyses, or interpretation of data; in the writing of the manuscript; or in the decision to publish the results.

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
