# Peer review of "Emission Factors for Biofuels and Coal Combustion in a Domestic Boiler of 18 kW"

_atmosphere, doi:10.3390/atmos10120771_

Round 1
Reviewer 1 Report
This manuscript discusses the emission factors for biofuels and coal combustion in a domestic boiler. The method of experimentation is suitable for the purpose. It includes a satisfactory and properly selected list of relevant references. However, I suggest that it should be accepted for publication after major revisions described below.
Introduction: The main aims of this work should be specified in detail. Moreover, CO2, SO2 and TC parameters were also studied.
Experiments: Only the numbers of the standard methods for references should be given in this section. Please specified them in the Reference section of this manuscript. More detailed description about and measurements of fuel characteristics (2.1) and PAHs (2.4) would be useful.
Results and discussions: How were the emission factors calculated in this work? It is missing from the text. Were the PAH molecular diagnostic ratios calculated from the average concentration data? It is not clear in this section or from the Experiments. How many PAH measurements were carried out?
Conclusions: Only, the most important results are listed in the present form. The most important findings should be summarized in this section.
Do not write one-sentence paragraphs in the text. Please check the full manuscript. Please correct the Table 5 and 6 English captions.
Author Response
Dear reviewer, we would like to thank you very much for your constructive reviews, below we present the answers
1.Introduction: The main aims of this work should be specified in detail. Moreover, CO2, SO2 and TC parameters were also studied.
The answer: The purpose of the work has been completed
Experiments: Only the numbers of the standard methods for references should be given in this section. Please specified them in the Reference section of this manuscript. More detailed description about and measurements of fuel characteristics (2.1) and PAHs (2.4) would be useful.The answer: the names of the reference methods are included in the references. The methods of fuel parameters determination, sampling and PAH determination are described in more detail.
3 Results and discussions: How were the emission factors calculated in this work? It is missing from the text. Were the PAH molecular diagnostic ratios calculated from the average concentration data? It is not clear in this section or from the Experiments. How many PAH measurements were carried out?
The answer: the relevant information can be found in the text
Conclusions: Only, the most important results are listed in the present form. The most important findings should be summarized in this section.The answer: The conclusions are completed
5 Do not write one-sentence paragraphs in the text. Please check the full manuscript. Please correct the Table 5 and 6 English captions.
The answer: Correction of the text with English correction has been done
Reviewer 2 Report
Issue 1
The work contains abbreviations, symbols and acronyms. Therefore, I suggest including a nomenclature section in the article.
Issue 2
The introduction is too general, in which generally presented data are well known and presented in the literature. However, there was no information on similar studies and their results presented by other authors.
Issue 3
The presentation of the measurement results in tables is hard to read and makes comparing them difficult. Presenting the results in the form of bar charts will be a much better solution.
Issue 4
The study was of an experimental nature, and therefore, an analysis of measurement uncertainties must be conducted. Measurement uncertainties should be presented in charts in the form of error bars or the values should be included in a separate table.
Author Response
Dear reviewer, we would like to thank you very much for your constructive reviews, below we present the answers
1.The work contains abbreviations, symbols and acronyms. Therefore, I suggest including a nomenclature section in the article.
The answer: a list of acronyms has been added
2. The introduction is too general, in which generally presented data are well known and presented in the literature. However, there was no information on similar studies and their results presented by other authors.
The answer The introduction has been extended
3. The presentation of the measurement results in tables is hard to read and makes comparing them difficult. Presenting the results in the form of bar charts will be a much better solution.
The answer: Part of the results presented in the form of tables have been transformed into figures
4. The study was of an experimental nature, and therefore, an analysis of measurement uncertainties must be conducted. Measurement uncertainties should be presented in charts in the form of error bars or the values should be included in a separate table.
The answer: Added uncertainty of measurement of concentrations of gaseous pollutants and added confidence interval for C, H, N, S and Cl determinations, OC, EC emission factors and PAHs molecular diagnostic ratio
Reviewer 3 Report
This is a kind of good study, but needs substantial modification before publication.
Line 9–19: In abstract, please try to include one or two more sentences including some results of the study. Expand PAH. Biomass combustion or biofuel combustion? Be consistent.
Line 22–24: Rephrase the sentence. Biofuel combustion emits substantial amount of CO, which by oxidation can contribute to CO2 level. It should be through modifying aerosol optical properties.
See some references on regional biomass burning
Pani et al (2018) Radiative response of biomass-burning aerosols over an urban atmosphere in northern peninsular Southeast Asia. Science of the Total Environment. 633, 892–911.
Tsay et al (2016) Satellite-surface perspectives of air quality and aerosol-cloud effects on the environment: An overview of 7-SEAS/BASELInE Aerosol and Air Quality Research. 16 (11), 2581–2602.
Add a more paragraph in introduction by describing the necessity and importance of this type of study. How this study will help the scientific community?
Line 43: “were combustion” what it means? Rephrase it with correct English.
Line 67–68: How the flue gas stream was sampled? And the gaseous pollutants were measured? Describe in details.
Line 75–77: Try to describe the analytical methods in more detail. What about the quality assured and quality control of data?
Line 208–219: Are the results obtained in this study comparable with any other studies?
Author Response
Dear reviewer, we would like to thank you very much for your constructive reviews, below we present the answers
Line 9–19: In abstract, please try to include one or two more sentences including some results of the study. Expand PAH. Biomass combustion or biofuel combustion? Be consistent.The answer: The abstract has been supplemented
Line 22–24: Rephrase the sentence. Biofuel combustion emits substantial amount of CO, which by oxidation can contribute to CO2 level. It should be through modifying aerosol optical properties.The answer: The modification has been made
See some references on regional biomass burningPani et al (2018) Radiative response of biomass-burning aerosols over an urban atmosphere in northern peninsular Southeast Asia. Science of the Total Environment. 633, 892–911.
Tsay et al (2016) Satellite-surface perspectives of air quality and aerosol-cloud effects on the environment: An overview of 7-SEAS/BASELInE Aerosol and Air Quality Research. 16 (11), 2581–2602.
The answer: Added references
Add a more paragraph in introduction by describing the necessity and importance of this type of study. How this study will help the scientific community?The answer: Chapter introduction extended
Line 43: “were combustion” what it means? Rephrase it with correct English.The answer: A linguistic correction has been made
Line 67–68: How the flue gas stream was sampled? And the gaseous pollutants were measured? Describe in details.The answer: The method of sampling and measurement is described
Line 75–77: Try to describe the analytical methods in more detail. What about the quality assured and quality control of data?The answer: The description of analytical methods has been extended
Line 208–219: Are the results obtained in this study comparable with any other studies?
The answer: added
Round 2
Reviewer 1 Report
Comments of reviewers have been incorporated into the manuscript. This version is well-written.
Author Response
Thank you for the information "Comments of reviewers have been incorporated into the manuscript. This version is well-written.
Reviewer 2 Report
I accept the modifications of the manuscript and responses provided by the authors.
Author Response
Thank you for the information "I accept the modifications of the manuscript and responses provided by the authors.